# Comparative Transcriptome Analysis Provided a New Insight into the Molecular Mechanisms of Epididymis Regulating Semen Volume in Drakes

**DOI:** 10.3390/ani12213023

**Published:** 2022-11-03

**Authors:** Xinyue Hu, Qingyuan Ouyang, Bincheng Tang, Xin Zhang, Jiwei Hu, Bo Hu, Shenqiang Hu, Liang Li, Hua He, Hehe Liu, Jiwen Wang

**Affiliations:** Farm Animal Genetic Resources Exploration and Innovation Key Laboratory of Sichuan Province, Sichuan Agricultural University, Chengdu 611130, China

**Keywords:** epididymis, avian sperm, seminal plasma, RNA-seq, bird

## Abstract

**Simple Summary:**

In recent years, some studies have found that epididymis could affect the reproductive performance of poultry. In drakes, the semen volume is regulated by epididymis, and the semen volume is the key to limit the ratio of male and female during artificial insemination. Therefore, the present study compared the histomorphology and transcriptome level differences in the epididymis of drakes with high- and low-volume semen. After bioinformatics analysis, nine differentially expressed genes (including *SLC18A2*, *SNAP25*, *CACNA1B*, *GABRG2*, *DRD3*, *CAMK2A*, *NR5A1*, and *STAR*) that up-regulated in the high-volume semen group and could regulate mutually were screened. The results reveal that the nine genes could affect the semen volume of drakes by regulating the synthesis of steroid hormones in epididymis. These results provide a new insight into the molecular mechanism regulating semen volume of drakes.

**Abstract:**

Semen volume is an important factor in artificial insemination (AI) of ducks. In drakes, seminal plasma that is produced by the epididymis determines the semen volume. However, the mechanism of epididymis regulating semen volume of drakes remains unclear. Therefore, the aim of the present study was to preliminarily reveal the mechanism regulating the semen volume through comparing the epididymal histomorphology and mRNA expression profiles between drakes with high-volume semen (HVS) and low-volume semen (LVS). Phenotypically, drakes in the HVS group produced more sperm than drakes in the LVS group. In addition, compared with the HVS group, the ductal square of ductuli conjugentes (DC) and dutus epididymidis (DE) in epididymis was significantly smaller in the LVS group, and the lumenal diameter and epithelial thickness of DC/DE were significantly shorter in the LVS group. In transcriptional regulation, 72 different expression genes (DEGs) were identified from the epididymis between HVS and LVS groups. Gene Ontology (GO) analysis indicated that the DEGs were mainly related to hormone secretion, neurotransmitter synthesis/transport, transmembrane signal transduction, transmembrane transporter activity, and nervous system development (*p* < 0.05). Kyoto Encyclopedia of Genes and Genomes (KEGG) functional enrichment analysis showed that the DEGs were significantly enriched in pathways associated with hormone and neurotransmitter transmission (*p* < 0.05). In addition, further analysis of the top five pathways enriched by KEGG, nine key candidate genes (including *SLC18A2*, *SNAP25*, *CACNA1B*, *GABRG2*, *DRD3*, *CAMK2A*, *NR5A1*, and *STAR*) were identified, which could play a crucial role in the formation of semen. These data provide new insights into the molecular mechanism regulating semen volume of drakes and make feasible the breeding of drakes by semen volume.

## 1. Introduction

Development of artificial insemination (AI) technology in poultry facilitates the progress of breeding male poultry. Male fertility is often measured by semen quality that can be further quantified into semen volume, sperm vitality, sperm motility, sperm concentration, acrosome integrity, and morphologically abnormal sperm [1,2]. Compared with chickens, AI technology in drakes is still relatively backward. This is mainly reflected in the unstable fertilization rate that is caused by the varying semen quality of drakes [3,4]. Scholars have uncovered some molecular mechanisms regulating sperm motility of male poultry in the past decade [5,6,7,8,9]. However, as an important influencing factor in the process of AI, semen volume has not attracted enough attention and needs further research.

Semen is the sum of sperm and seminal plasma. Seminal plasma in mammals accounts for 80 to 90% of semen [10,11]. In poultry, seminal plasma is mainly produced by epididymis [12,13]. Unlike mammals, which divided the epididymis into caput, corpus, and cauda parts [14], the epididymis of poultry is divided into appendix epididymidis (Ae) and main epididymis (Em) parts. Additionally, there are two kinds of ducts, called ductuli conjugentes (DC) and dutus epididymidis (DE), in the epididymis of poultry [15,16]. The lumens of DC and DE are accompanied by a layer of epithelial tissue that is the main part of secreting seminal plasma [12,13,14,15,16]. Distinctively, after sexual maturity, the DC of male poultry is significantly smaller than the DE, and microvilli (derived from brush cells) presenting in DE epithelium are dense, whereas microvilli are sparse in DC epithelium [15,16]. The epididymis of mammals can regulate sperm maturation while, for a long time, the epididymis of poultry was considered irrelevant to sperm maturation [17,18]. In recent years, a few studies showed that epididymal dysfunction could cause sterility in cocks [19,20]; these indicated that epididymis may play an important role in the reproductive performance of poultry. Nevertheless, it is still unknown whether epididymis regulates reproductive performance of poultry by semen volume.

Ribonucleic acid sequencing (RNA-seq) technology is favored by researchers because of its high efficiency and rapidness. Lots of studies found some molecular mechanisms of gonadal development and spermatogenesis by using RNA-seq technology to research reproductive tract in mammals [21,22]. Recently, RNA-seq technology has also been used to explore libido and external genital development of poultry [23,24]. However, the mechanism of epididymis regulating semen volume in drakes remains unclear. Therefore, the aim of the present study was to compare the transcriptome profiles of the epididymis between drakes with high-volume semen (HVS) and low-volume semen (LVS) by RNA-seq technology. These data will help to reveal the molecular mechanism that regulates the semen volume of drakes.

## 2. Materials and Methods

### 2.1. Ethics Approval and Consent to Participate

All experimental procedures were approved by the Institutional Animal Care and Use Committee (IACUC) of Sichuan Agricultural University (Chengdu Campus, Chengdu, China), and the permit number is DKY20170913.

### 2.2. Management of Experimental Drakes

A total of 60 drakes that hatched in the same batch were kept in the Sichuan Agricultural University Waterfowls Breeding Farm (Ya’an Campus, Ya’an, China). At 140 days of age, all drakes were reared in single cages at natural temperature and were provided with natural light from 6:00 to 23:00 daily. Feed and water were available at random. Drakes were trained with abdominal massage technique at 190 days of age until semen was successfully collected from each one. Abdominal massage training was performed by a skilled technician.

### 2.3. Semen Quality Analysis of the Drakes

In the present study, semen quality of the 60 drakes was assessed three times over a three-day interval, at their 210 and 364 days of age (at 364 days of age, nine drakes had been sacrificed for other study, so their semen quality parameters were not assessed). The measured methods of semen quality were adjusted according to previous reports [25,26], and the detailed procedures were as follows: (1) A professional used a 1.5 mL syringe to measure the semen volume after collecting the semen into a 150 mL high beaker (Shubo, Chengdu, China) by abdominal massage technique; (2) After the semen was diluted at a ratio of 1:20 with 0.9% stroke-physiological saline solution (NaCl), ten µL of the diluted semen (0.9% NaCl, 20×) was added on a slide which was preheated at 37 °C to measure sperm motility. The natural motions of at least 200 sperm on the slide were observed by using a phase contrast microscope (Olympus, Tokyo, Japan) at 400×. Sperm that moving in a straight line were considered to possess sperm motility; (3) Semen was mixed with 1% trypan blue solution (Sangon, Shanghai, China) in equal proportion and incubated in a metal bath (Allshneg, Hangzhou, China) for 15 min at 37 °C, ten µL of the above mixture was added on a slide which was preheated at 37 °C to measure sperm viability. The color of at least 200 sperm on the slide was observed by using a phase contrast microscope at 400×. Colorless sperm were considered alive, and sperm that stained blue were considered sacrificial; (4) Smeared 20 µL diluted semen (0.9% NaCl, 20×) onto one slide, allowed to air dry and stained the slide in 0.5% gentian violet solution (Sangon, Shanghai, China) for three minutes, and then phase contrast microscope was used to examine the morphology of at least 300 sperm at 400×. Sperm that were linear from head to tail were considered normal, while others were considered morphologically abnormal; (5) Smeared ten µL diluted semen (0.9% NaCl, 20×) onto one slide, allowed to air dry and stained the slide in phosphate buffered saline solution (Sangon, Shanghai, China) containing 2% glutaraldehyde (Sangong, Shanghai, China) for 15 min, and then transferred to 5% aniline blue solution (Sangon, Shanghai, China) for five minutes. Acrosome integrity of at least 100 sperm in the slide was observed under a phase contrast microscope at 1000× (Oil immersion); (6) After the semen was diluted at a ratio of 1:400 with 3% sterile NaCl solution, ten µL of the diluted semen (3% NaCl, 400×) was added on a hemocytometer to measure sperm concentration by using a phase contrast microscope at 400×. Total sperm number of each semen sample was calculated through the following equation: Total sperm number = semen volume (mL) × sperm concentration (×10^9^/mL).

### 2.4. Sample Collection

According to the semen volume both at 210 and 364 days of age, drakes with the top 10 semen volume were divided into the HVS group and drakes with the bottom 10 semen volume were divided into the LVS group. Meanwhile, the 20 drakes were weighed at 376 days of age. Seven healthy drakes with similar body weight were screened in the HVS and LVS groups, respectively, for slaughter. Drakes were sacrificed by carotid artery bloodletting after carbon dioxide anesthesia. After drakes were sacrificed, the whole left epididymides were removed immediately. The epididymides of four drakes in each group were fixed in 4% paraformaldehyde (Beyotime, Shanghai, China); other epididymides were rapidly frozen by liquid nitrogen and stored at −80 °C until RNA isolation.

### 2.5. Histological Observation

After fixing the epididymides in 4% paraformaldehyde for 24 h, the Em of every epididymis was trimmed to appropriate size and dehydrated in 75, 85, 95%, and absolute ethanol, respectively. Then, the Em was embedded in paraffin. Two pieces of tissue with a thickness of 4 μm were cut from each Em using a rotary microtome (Leica, Munich, Germany) to make sections. These sections were stained with hematoxylin-eosin and photographed by mathematical pathological scanner (Olympus, Tokyo, Japan). Image-Pro plus 6.0 software (Media Cybernetics, MD, USA) was used to calculate the ductal square, lumenal diameter, and epithelial thickness of the biggest DC/DE and nine DCs/DEs surrounding them. Among, the duct consists of epithelium and lumen.

### 2.6. RNA-seq and Bioinformatics Analysis

Total RNA from the epididymis of three individuals in each group was extracted by using RNeasy Mini Kit (Qiagen, Beijing, China). NanoDrop 2000 Microultraviolet Spectrophotometer (Thermo Fisher Scientific, Wilmington, NC, USA) was used to determine RNA concentration and Agilent 2100 Bioanalyzer (Agilent Technologies, Santa Clara, CA, USA) was used to detect RNA integrity. RNA samples were reversely transcribed to produce cDNA libraries. The cDNA libraries were sequenced by Majorbio Co., Ltd. (Shanghai, China) using Nova-PE150 (Illumina, Sandiego, CA, USA) system. The raw sequencing data obtained in the present study can be found in the National Center for Biotechnology Information (NCBI) and the Biological Project ID is PRJNA791523 (https://submit.ncbi.nlm.nih.gov/subs/sra/SUB10824384/overview accessed on 22 June 2022). FastQC software (version 0.11.9) was utilized to obtain the clean reads. Clean reads were mapped to the reference genome of duck (GCA_015476345.1) by HISAT2 (version 2.2.1) software to obtain sequencing alignment/mapping (SAM) files [27]. SAM files were converted to binary alignment/mapping (BAM) files and sorted through SAMtools (version 1.6.0) software [28]. FeaturCounts (version 1.6.0) software was used to acquire the expression levels of every transcript [29]. Subsequently, the counts of each transcript were standardized to transcripts per million (TPM) by GenomicFeatures (version 1.46.3) package. DESeq2 (version 1.34.0) package was used to identify the different expression genes (DEGs) and the criteria of screening were |log2Foldchange| > 1 and *p*_adjust < 0.05 [30]. KOBAS (version 3.0) online software was used to analyze the Gene Ontology (GO) functions and the Kyoto Encyclopedia of Genes and Genomes (KEGG) functions [31]. The KEGG database was used to determine the relationships between KEGG pathways. The STRING 11.5 database was employed to identify the relationship between the DEGs identified in the present study. All the network visualization was performed using Cytoscape (version 3.7.1) software.

### 2.7. Quantitative Reverse Transcription-PCR (RT-qPCR) Validation

In order to prove the reliability and repeatability of sequencing data, nine DEGs (including *STAR*, *SLC18A2*, *CAMK2A*, *DRD3*, *CACNA1B*, *CDK5R2*, *NR5A1*, *GABRG2*, and *SANP25*) that were up-regulated in the HVS group were selected for RT-qPCR verification. The HiScript^®^ III RT Kit (Vazyme, Nanjing, China) was used to convert mRNA into cDNA. Primer 5.0 software was used to design primers (Table 1) spanning exon. Expression levels of the nine DEGs were detected by SYBR Green method in the Bio-Rad CFX96 real-time PCR system (Bio-Rad, Hercules, CA, USA). The *GAPDH* and *β-actin* were used as housekeeping genes and three technical replicates were performed. Systems of qPCR reaction were as follows: SYBR Green PCR SuperMix (Vazyme, Nanjing, China) was 10 µL, both PCR forward and reverse primers (10 µM) were 0.4 µL, ddH2O was 7.2 µL, cDNA was 2 µL. The 2^−ΔΔCT^ method was used to analyze RT-qPCR data [32].

### 2.8. Statistical Analysis

SPSS 27.0 (IBM, Chicago, TL, USA) software was used for statistical analysis. The means of semen volume, sperm motility, sperm vitality, morphological abnormal sperm, acrosome integrity, sperm concentration, total sperm number, ductal square, lumenal diameter, and epithelial thickness between HVS and LVS groups were subjected to ANCOVA testing with body weight as a covariate, and *t*-test was utilized to analyze the significance between the two groups. Pictures were plotted by Graph Pad Prism (version 5.0) software and results in the pictures were expressed as mean ± standard deviation. The criterion of statistically significance was considered at *p* < 0.05.

## 3. Results

### 3.1. Semen Quality Parameters between HVS and LVS Groups

As shown in Table 2, when body weight was used as a covariate, compared with the LVS group, drakes in HVS group had higher semen volume (*p* < 0.01) and total sperm number (*p* < 0.01). Nevertheless, other semen quality parameters, including sperm viability, sperm motility, morphological abnormal sperm, acrosome integrity, and sperm concentration, were not statistically significant between the two groups.

### 3.2. Morphological Differences in the Epididymis between HVS and LVS Groups

There were two kinds of ducts, called DC and DE, in the Em of drakes; both DC and DE had a similar layer of epithelial tissue. The difference was that microvilli in the epithelial tissue of DE were closely arranged while microvilli in the epithelial tissue of DC were sparsely arranged (Figure 1A). In addition, compared with the HVS group, the ductal square of DC/DE was significantly smaller in the drakes of LVS group (*p* < 0.01, Figure 1B), and the lumenal diameter and epithelial thickness of DC/DE were significantly shorter in the drakes of LVS group (*p* < 0.01, Figure 1C,D).

### 3.3. Overview of RNA-seq Data

As shown in Appendix A, a total of 311,039,855 raw reads were obtained from six samples after RNA-seq, and 308,349,550 clean reads were obtained through strict filtering. The Q20 (percentage of reads with a Phred quality value > 0) and Q30 (percentage of reads with a Phred quality value > 30) of all samples were above 98% and 95%, respectively. The mapping rate of six samples ranged from 90.31 to 91.38%. In addition, 72 DEGs were identified from epididymis between HVS and LVS groups, 56 of DEGs were up-regulated and 16 were down-regulated (Figure 2A). The heat map (Figure 2B) showed the different gene expression patterns in the epididymis between two groups.

### 3.4. Functional Analysis of DEGs between HVS and LVS Groups

The DEGs identified from epididymis of HVS and LVS groups were annotated with GO database into the biological process (BP), cellular component (CC) and molecular function (MF) categories, and the top ten GO terms enriched in BP, CC and MF, respectively, were shown in Figure 3A. As shown in Appendix A, DEGs annotated to BP were mainly associated with hormone secretion, neurotransmitter synthesis/transport, transmembrane signaling, transmembrane transporter activity, and nervous system development; DEGs annotated to CC were mainly associated with transmembrane transporter activity and nervous system development; DEGs annotated to MF were mainly associated with hormone secretion, transmembrane signaling, and transmembrane transporter activity. In addition, the DEGs were enriched in 19 KEGG pathways (*p* < 0.05, Figure 3B) and the 19 pathways were related with hormone (e.g., Cushing syndrome, insulin secretion, and renin-angiotensin system) and neurotransmitter transmission (e.g., neuroactive ligand-receptor interaction, dopaminergic synapse, synaptic vesicle cycle, and cholinergic synapse).

### 3.5. Network Analysis and RT-qPCR Validation of the DEGs Involved in Regulating Semen Volume

The results of PPI (protein-protein interaction) analysis were shown in Appendix A. In order to identify the key genes regulating semen volume, the distribution of DEGs with node number greater than ten in KEGG pathways were further observed, and it was found that 70% of these DEGs were enriched in the top five pathways that existed crosstalk. Subsequently, nine DEGs (including *SLC18A2*, *SNAP25*, *CACNA1B*, *GABRG2*, *DRD3*, *CAMK2A*, *NR5A1*, and *STAR*) enriched in the five pathways were selected to construct an interaction network diagram (Figure 4A). Moreover, the expression levels of the nine key genes that generated from RT-qPCR were similar to the RNA-seq results (Figure 4B), indicated the reliability of RNA-seq results.

## 4. Discussion

In production practice, the semen volume of male poultry is related to economic benefits. In the present study, the total sperm number in the HVS group was significantly higher than that in the LVS group, and other semen quality parameters (including sperm viability, sperm motility, morphological abnormal sperm, acrosome integrity, and sperm concentration) did not have statistical significance between the two groups, indicating that the semen volume could positively affect the total sperm number of drakes and could affect the fertility of drakes by increasing the male-female ratio. Previous studies showed that the seminal plasma of poultry is mainly produced by DC and DE in the epididymis [12,13,14,15,16]. In the present study, compared with the LVS group, the development of DC and DE was found to be better in the drakes of HVS group. Meanwhile, previous research showed that seminal plasma could enhance the sperm motility of poultry [33]. The finding that sperm motility in the drakes of HVS group was higher than in the LVS group of the present study also confirmed this point. The important role of sperm motility in the reproduction of poultry has been repeatedly recognized [34,35,36]. These results suggested that epididymis could affect the reproductive performance of drakes by regulating semen volume.

In the present study, 72 DEGs were identified in the epididymis between HVS and LVS groups, suggesting that semen volume of drakes could be mainly regulated by a few key genes. The biological implications of the 72 DEGs were further confirmed through function analyses. In GO analysis, most of DEGs were significantly enriched in the GO terms associated with hormone secretion, neurotransmitter synthesis/transport, transmembrane signaling, transmembrane transporter activity, and nervous system development, suggesting that semen volume could be regulated by both hormones and nerves. In KEGG analysis, most of DEGs were significantly enriched in neuroactive ligand-receptor interaction, dopaminergic synapse, synaptic vesicle cycle, cholinergic synapse, and Cushing syndrome pathways. Studies have shown that the neuroactive ligand-receptor interaction [37], dopaminergic synapse [38], synaptic vesicle cycle [39], cholinergic synapse [40], and Cushing syndrome pathways [41] could play important roles in regulating reproductive performance. Further analysis showed that the five pathways have crosstalk, and DEGs enriched in the five pathways were up-regulated in HVS group, suggesting that these DEGs could play an important role in the epididymis regulating semen volume.

When the present study attempted to explain how DEGs that enriched in the five pathways regulated the semen volume of drakes, researchers noticed that nine DEGs (including *SLC18A2*, *SNAP25*, *CACNA1B*, *GABRG2*, *DRD3*, *CAMK2A*, *NR5A1*, and *STAR*) have mutual regulating relationships (Figure 4A). The gene of *CDK5R2* could promote brain development and participate in dopamine signaling by encoding Cdk5 (cyclin-dependent kinase) protein [42,43]. A previous study has found that the brain could secrete DA (dopamine) and CRH (corticotropin-releasing hormone) [44]. Dopamine could enter synaptic vesicles with the assistance of *SLC18A2* and bind to SNARE (soluble N-ethylmaleimide-sensitive factor attachment protein receptor) complexes which were composed of *SNAP25* and located in the presynaptic membrane [45,46]. Subsequently, the DA (dopamine) was released into the interstitial fluid with the assistance of *CACNA1B* [47], whereas synaptic vesicles were blended with ER (endoplasmic reticulum) under the help of *GABRG2* [48,49]. Besides, CRH (corticotropin-releasing hormone) entered the interstitial fluid directly through pituitary portal system [50]. Then, the dissociative DA (dopamine) and CRH (corticotropin-releasing hormone) in interstitial fluid could combine with specific receptors that are located in the postsynaptic membrane to enhance the expression level of *CAMK2A* [51,52,53]. Noticeably, Kimura found that CRH (corticotropin-releasing hormone) overexpression did not affect the content of plasmatic ACTH (adrenocorticotropic hormone) in *CAMK2A* knockout mice [54]. However, CRH (corticotropin-releasing hormone) controlled the production of ACTH (adrenocorticotropic hormone) [55]. These results suggested that *CAMK2A* could regulate ACTH (adrenocorticotropic hormone) synthesis. In addition, studies have shown that ACTH (adrenocorticotropic hormone) regulated the synthesis of steroid hormones with the assistance of *NR5A1* and *STAR* [56,57], and steroid hormones could participate in regulating organ development and reproductive performance [58,59]. Interestingly, the present study found that compared with the HVS group, the ductal square of DC/DE in epididymis was significantly smaller in the LVS group, and the lumenal diameter and epithelial thickness of DC/DE were significantly shorter in the LVS group, suggesting that the epididymis of drakes in HVS group developed better. All the results obtained in the present study suggested that the synthesis of steroid hormones in the epididymis may contribute to the difference in semen volume of drakes by regulating the development of epididymis.

## 5. Conclusions

In conclusion, the present study both found that drakes with high-volume semen produced more sperm, and constructed the first mRNA expression profile in the epididymis of drakes with high- and low-volume semen. In addition, bioinformatics analysis suggested that nine DEGs (including *SLC18A2*, *SNAP25*, *CACNA1B*, *GABRG2*, *DRD3*, *CAMK2A*, *NR5A1*, and *STAR*) identified from neuroactive ligand-receptor interaction, dopaminergic synapse, synaptic vesicle cycle, cholinergic synapse, and Cushing syndrome pathways were crucial for the semen volume of drakes. The nine DEGs could affect the semen volume of drakes by regulating the synthesis of steroid hormones in the epididymis. These results provide a new insight into the molecular mechanism regulating the semen volume of drakes.

## Figures and Tables

**Figure 1 animals-12-03023-f001:**
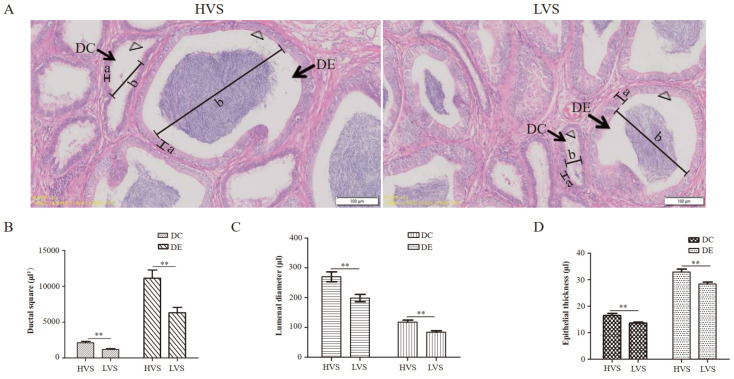
Histological differences of epididymis between HVS and LVS groups: (**A**) morphological differences of epididymis in drakes with different semen volume (1000×); (**B**) differences in ductal square of DC and DE between HVS and LVS groups; (**C**) differences in lumenal diameter of DC and DE between HVS and LVS groups; (**D**) differences in epithelial thickness of DC and DE between HVS and LVS groups. “DC” means ductuli conjugentes; “DE” means dutus epididymidis; “a” means the epithelial thickness; “b” means the lumenal diameter; “Δ” means microvilli; “HVS” means high-volume semen; “LVS” means low-volume semen; “**” means *p* < 0.01.

**Figure 2 animals-12-03023-f002:**
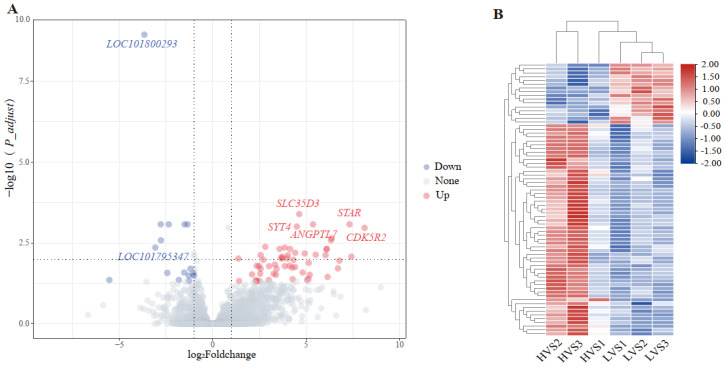
Volcano plot (**A**) and heat map (**B**) of DEGs that were screened from epididymis of drakes with different semen volume. Each dot in volcano plot represents one gene, red dots represent up-regulated genes, blue dots represent down-regulated genes, and gray dots represent unchanged genes. “HVS” means high-volume semen; “LVS” means low-volume semen.

**Figure 3 animals-12-03023-f003:**
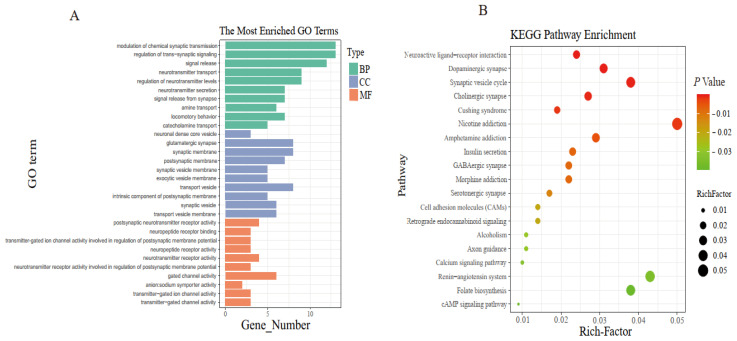
(**A**) Gene Ontology (GO) terms enriched by DEGs and (**B**) Kyoto Encyclopedia of Genes and Genomes (KEGG) pathways enriched by DEGs. “BP” means biological process; “CC” means cellular component; “MF” means molecular function. The Rich-Factor is the ratio of the number of DEGs in the pathway and the total number of genes in the pathway.

**Figure 4 animals-12-03023-f004:**
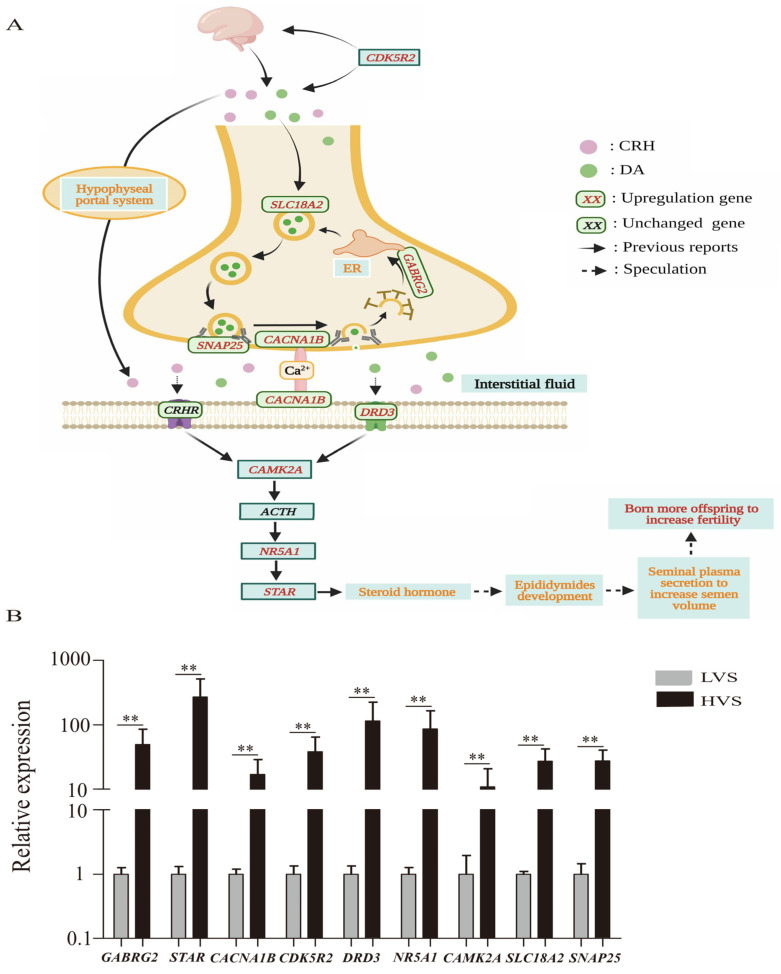
(**A**) Regulation network involved in semen volume of drakes and (**B**) the expression levels of *STAR*, *CACNA1B*, *CDK5R2*, *DRD3*, *NR5A1*, *CAMK2A*, *SLC18A2*, *SANP25*, and *GABRG2* that were detected from epididymis of drakes with different semen volume by RT-qPCR. “DA” means dopamine; “CRH” means corticotrophin-releasing hormone; “ER” means endoplasmic reticulum; “HVS” means high-volume semen; “LVS” means low-volume semen; “**” means *p* < 0.01.

**Table 1 animals-12-03023-t001:** Primers used in this study.

Primer Name	Sequence (5′-3′)	Product Length (bp)
*STAR-F*	GCAGAAGATTGGGAAAGACACG	246
*STAR-R*	AACTTGGTTTGCGAGGGGTC	
*CACNA1B-F*	GATTGACTTGGGGCTGTTACTG	144
*CACNA1B-R*	GTGTTGATGTCCTTTCCTTTGG	
*CAMK2A-F*	GCGTGAAGAAAAGAAAGTCCAG	187
*CAMK2A-R*	ATCCCAGGGTCACACATCTTC	
*CDK5R2-F*	CTGCCTCTATCTCGCCTACTCC	290
*CDK5R2-R*	AATCCGTCAGTCCTTGTCCTCC	
*DRD3-F*	GATTGTGATAGTCTGGATGCTGGC	118
*DRD3-R*	GGAGTAGATGACGAAAATAGGGTTG	
*GABRG2-F*	TCCTCCAAATCCAACAAGC	169
*GABRG2-R*	GGGAGTCAAAACCCAAGTCT	
*NR5A1-F*	CAGACCCTCTTCTCCATCGTG	363
*NR5A1-R*	CTTAGCCAGCGTGTGGTTCTC	
*SLC18A2-F*	AGTCAGAAGGGGACACCATTA	172
*SLC18A2-R*	GAAGAAAAGCAACGCCAAG	
*SNAP25-F*	AATCCCTTGAGAGCACCCG	312
*SNAP25-R*	CCATCTGCTCCCGTTCATCTA	
*GAPDH-F*	GTCTCTGTCGTGGACCTGAC	113
*GAPDH-R*	GTGTATGCCAGGATGCCCTT	
*β-actin-F*	GCTATGTCGCCCTGGATTTC	168
*β-actin-R*	CACAGGACTCCATACCCAAGAA	

Note: genes are shown in italics.

**Table 2 animals-12-03023-t002:** Semen quality parameters of drakes in high- and low-volume semen groups.

Semen Variables	HVS (*n* = 10)	LVS (*n* = 10)	*p*-Value
Semen volume (mL)	0.70 ± 0.03	0.21 ± 0.03	<0.01 **
Sperm vitality (%)	94.40 ± 0.20	94.30 ± 0.20	0.653
Sperm motility (%)	91.30 ± 0.60	90.70 ± 0.60	0.444
Morphological abnormal sperm (%)	7.10 ± 0.90	6.80 ± 0.90	0.803
Acrosome integrity (%)	94.00 ± 0.90	95.60 ± 0.90	0.249
Sperm concentration (10^9^/mL)	3.61 ± 0.30	3.83 ± 0.30	0.597
Total sperm number (10^9^)	2.55 ± 0.22	0.81 ± 0.22	<0.01 **

Note: statistically significant at *p* < 0.05. “HVS” means high-volume semen; “LVS” means low-volume semen; “**” means *p* < 0.01.

## Data Availability

The original sequencing data for this study can be found in the Sequence Read Archive (https://submit.ncbi.nlm.nih.gov/subs/sra/SUB10824384/overview accessed on 22 June 2022) at NCBI with the BioProject ID: PRJNA791523.

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
