# Peer review of "Comparative Transcriptome Analysis Provided a New Insight into the Molecular Mechanisms of Epididymis Regulating Semen Volume in Drakes"

_animals, 2022, doi:10.3390/ani12213023_

Round 1

Reviewer 1 Report

This is an interesting study that investigates the histomorphology and transcriptome level differences in the epididymis of drakes presenting high and low volume semen. 

Tittle - Adequate

Summary and abstract - Ok

Keywords - Authors are advised to provide keywords different from that previously reported in the tittle. Some suggestions: bird, avian sperm, seminal plasma, etc.

Introduction - This section is well written and clearly presents some justifications for the execution of the research.

Methods - With relation to the number of individuals, authors used only the 10 individuals presenting highest volume and other 10 individuals presenting low semen volume, isn't it? And the other 40 individuals, were they discarded?

- In the birds, is not the epididymis divided in caput, corpus and cauda? If yes, should not histomorphological and transcriptome-related analyzes be performed separately from each region of the epididymis?

Results - These are well presented, in general.

Discussions - If all the sperm characteristics are similar between individuals presenting high or low volume, would the fertility patterns be similar? How only volume could affect fertility? This is an interesting point to be addressed in discussion.

- Authors did not discuss their findings related to histomorphology. Please add it.

- Discussions related to transcriptome analysis are ok.

Conclusions - OK

References - OK

Reviewer 2 Report

This study is interesting. The authors compared the transcriptome of epididymis between the top 20% and bottom 20% semen volume in drakes. My comments are as follows:

Line 50 delete “been”

Line 54 change “mammalian is” to “mammals”

Line 92-93, Did semen volume and total sperm number correlate with their age or weight?

Has it been analyzed? Did abdominal massage ensure that all semen can be extracted?

Line 124-126, a total of 60 drakes, the top 20% should be 12 drakes, not 10 drakes. How many male drakes were used in this study? How the semen volume was determined? The authors just used the semen volume, why not use the total sperm number? A larger volume of semen does not necessarily mean that there are more viable sperm in it.

Line 159, It's better to replace “expression levels” with “counts”.

Line 163, “log2Foldchange”

Line 180-187, why not consider age and body weight of the drakes as covariables?

Line 224, Figure 2A, The ordinate of A is –logP_adjust (do you want used –log10P_adjust)? Figure 2B, DEGs not have the same expression trend as HVS2 and HVS3 in HVS1, how about the semen quality parameters of HVS1?

Line 231, It's better to replace “GO pathways” with “GO terms”.

Line 244, Figure 3A, The x-axis shows an incomplete gene_ number, please improve the clarity of the picture.

Line 254, Figure 4A, the interaction network diagram is interesting, but it is one-sided to select only the genes enriched top5 KEGG pathways as key genes. In terms of this mechanism diagram, steroid hormones are the main factors affecting semen volume, STAR gene is an important regulator of steroid hormone synthesis, but the effects of steroid hormones are numerous. The authors can use others methods to search for more genes potentially regulating semen volume, such as PPI network or GESA analysis.

Line 270, change “mainly produced” to “is mainly produced”

Line 272, change “was found better” to “was found to be better”

Line 280, 285, 292, change “suggested” to “suggesting”

Line 300, change “Previous study” to A previous study”

Discussion: It is suggested to discuss the relationships between semen volume and total sperm number. As we known, spermatozoa are generated by testes, while epididymis only generates seminal plasma, and spermatozoa are stored in the epididymis. Is the seminal plasma production of epididymis determined by the spermatozoa production in the testis? Why did the authors not analyze the gene expression profiles in testes at the same time?

Author Response

Point 1: Line 50: delete “been”.

Response 1: We apologize for the mistakes in the manuscript and also carefully checked the entire manuscript for typographic, grammatical and formatting errors. As you suggested, the “been” has been deleted.

Point 2: Line 54: change “mammalian is” to “mammals”.

Response 2: Done as requested.

Point 3: Line 92-93: Did semen volume and total sperm number correlate with their age or weight? Has it been analyzed? Did abdominal massage ensure that all semen can be extracted?

Response 3: Thank you for pointing out this problem. Previously, we had analyzed the correlation of semen volume and total sperm number with age/weight of drakes at their 210 and 364 days of age (both ages are sexually mature). Results are as follows: There was no significant (P > 0.05) correlation between the age of drakes and the sperm number (r = 0.156) and semen volume (r = 0.183), which was consistent with the results previously reported by Dorado (Dorado, J.; Acha, D.; Gálvez, M.J.; Ortiz, I.; Carrasco, J.J.; Díaz, B.; Gómez, A.V.; Calero, C.R.; Hidalgo, M. Sperm motility patterns in Andalusian donkey semen: effects of body weight, age, and semen quality. Theriogenology 2013, 79(7):1100-1109, doi:10.1016/j.theriogenology.2013.02.006.). The correlations between the weight of drakes and their semen volume (r = 0.260) and total sperm number (r = 0.038) were not significant (P > 0.05) too, which was consistent with Zhang's reports that the weight was positively correlated with semen volume and total sperm number during sexual maturity in chicken (Zhang, X.; Berry, W.D.; McDaniel, G.R.; Roland, D.A.; Liu, P.; Calvert, C.; Wilhite, R. Body weight and semen production of broiler breeder males as influenced by crude protein levels and feeding regimens during rearing. Poult Sci 1999, 78(2):190-196, doi:10.1093/ps/78.2.190.). In order to keep the length of the article manageable, we did not show this part of the results in the main text. Furthermore, with the development of artificial insemination of poultry, the method of abdominal massage for semen collection has been widely used in all kinds of poultry and some wild birds. Łukaszewicz also found that the semen of grouse collected by abdominal massage had a higher sperm density compared with the dummy female method, while other semen parameters were similar in both methods (Łukaszewicz, E. T.; Kowalczyk, A. M.; Rzońca, Z. Comparative Examination of Capercaillie Behaviour Responses and Semen Quality to Two Methods of Semen Collection. PLoS One 2015, 10(9):e0138415, doi:10.1371/journal.pone.0138415.). These results showed that abdominal massage is an effective way to collect semen. In addition, the collection of each semen sample in this study was carried out by a same technician at the same pace to avoid some unnecessary influencing factors.

Point 4: Line 124-126: a total of 60 drakes, the top 20% should be 12 drakes, not 10 drakes. How many male drakes were used in this study? How the semen volume was determined? The authors just used the semen volume, why not use the total sperm number? A larger volume of semen does not necessarily mean that there are more viable sperm in it.

Response 4: Thanks for your suggestion. In this study, the semen quality of 60 drakes was evaluated at their 210 days of age. However, at 364 days of age, nine drakes had been sacrificed for other experiment, so semen quality was evaluated only for the remaining 51 drakes. Subsequently, we divided these 51 drakes into high-volume semen (n = 10) and low-volume semen (n = 10) groups based on their semen volume at 210 and 364 days of age. According to your suggestion, we have made some modifications in “Materials and Methods”. Please see the changes in lines 94-95 and 126-128 of the revised manuscript. In addition, as described in the “Semen Quality Analysis of the Drakes” in our article, we used a 1.5 ml syringe to measure the semen volume after collecting the semen into a 150 ml high beaker (lines 97-99). The main purpose of this study was to reveal the mechanism regulating the semen volume of drakes. Whether there is a positive regulatory relationship between the semen volume and the total sperm number will be discussed in subsequent studies.

Point 5: Line 159: It's better to replace “expression levels” with “counts”.

Response 5: Done as requested.

Point 6: Line 163: “log2Foldchange”.

Response 6: Done as requested.

Point 7: Line 180-187: why not consider age and body weight of the drakes as covariables?

Response 7: According to your suggestion, we used weight (age was not considered because only two stages, 210 and 364 days of age, were involved in this study, making age equivalent to a categorical variable.) as a covariate to analyze the semen quality parameters of drakes in the high- and low-volume semen groups. The results showed that when weight was used as a covariate, semen volume (P < 0.01) and total sperm number (P < 0.01) in the high-volume semen group were significantly higher than those in the low-volume semen group, while other semen parameters (P > 0.05) were still not statistically significant in the two groups. We have changed this information in the article; please see the changes in lines 188-192 and 197-202 of the revised manuscript.

Point 8: Line 224: Figure 2A, The ordinate of A is –logP_adjust (do you want used –log10P_adjust)? Figure 2B, DEGs not have the same expression trend as HVS2 and HVS3 in HVS1, how about the semen quality parameters of HVS1?

Response 8: As you suggested, we have changed “–logP_adjust” to “–log10(P_adjust)” in the Figure 2A (Line 232). In addition, the semen quality parameters of HSV1 were similar to those of HVS2 and HVS3. The expression trend of DEGs in HVS1 was slightly different from HVS2 and HVS3, which may be caused by cirrhosis in HVS1. Moreover, compared with the LVS group, the expression levels of DEGs in HSV1 were more similar to HVS2 and HVS3.

Point 9: Line 231: It's better to replace “GO pathways” with “GO terms”.

Response 9: Done as requested.

Point 10: Line 244: Figure 3A, the x-axis shows an incomplete gene-number; please improve the clarity of the picture.

Response 10: Done as requested.

Point 11: Line 254: Figure 4A, the interaction network diagram is interesting, but it is one-sided to select only the genes enriched top 5 KEGG pathways as key genes. In terms of this mechanism diagram, steroid hormones are the main factors affecting semen volume, STAR gene is an important regulator of steroid hormone synthesis, but the effects of steroid hormones are numerous. The authors can use others methods to search for more genes potentially regulating semen volume, such as PPI network or GESA analysis.

Response 11: Thank you for your suggestion. We had conducted PPI analysis on DEGs previously, and our PPI analysis results showed that about 70% DEGs (including SNAP25, CDK5R2, GABRG2, CAMK2A, CACNA1B, SLC18A2, and DRD3) with nodes greater than 10 were enriched in the top 5 KEGG pathways. Therefore, we selected the DEGs enriched in the top 5 KEGG pathways to draw the regulatory mechanism plot. We have uploaded this result to the supplementary; please check it in the “Supplementary Figure S1”. In addition, among 19 KEGG pathways enriched by DEGs in this study, there was no condition that up-regulated genes and down-regulated genes were enriched into one pathway at the same time, which made GESA analysis was redundant.

Point 12: Line 270: change “mainly produced” to “is mainly produced”.

Response 12: Done as requested.

Point 13: Line 272: change “was found better” to “was found to be better”.

Response 13: Done as requested.

Point 14: Line 280, 285, 292: change “suggested” to “suggesting”.

Response 14: Done as requested.

Point 15: Line 300: change “Previous study” to “A previous study”.

Response 15: Done as requested.

Point 16: Discussion: It is suggested to discuss the relationships between semen volume and total sperm number. As we known, spermatozoa are generated by testes, while epididymis only generates seminal plasma, and spermatozoa are stored in the epididymis. Is the seminal plasma production of epididymis determined by the spermatozoa production in the testis? Why did the authors not analyze the gene expression profiles in testes at the same time?

Response 16: As your suggestion, we have added some discussions about the relationship between semen volume and total sperm number; please see the changes in lines 275-280 of the revised manuscript. In addition, seminal plasma is the main component of semen, and our results showed that the total sperm number of drakes in high-volume semen group was significantly higher than that of drakes in low-volume semen group, indicating that there should be a positive effect between seminal plasma volume and sperm number. We will consider carrying out corresponding research on this issue in the future. Finally, the role of testis in avian reproduction has been extensively studied and its molecular regulatory mechanisms are well understood. However,the role of epididymis in the reproductive process of poultry is still relatively blank, which make it is an interesting scientific problem to investigate the mechanism of epididymis regulating semen volume in drakes.

Finally, we really appreciate all your comments and suggestions. Thanks very much for taking your time to review this manuscript. Those comments are all valuable and very helpful for revising and improving our paper. We have revised the manuscript accordingly, and our point-by-point responses are presented above.

Round 2

Reviewer 2 Report

After the author's revision, the manuscript has been greatly improved.  

The authors addressed my concerns. Thanks!